# Acute Physical Activity, Executive Function, and Attention Performance in Children with Attention-Deficit Hyperactivity Disorder and Typically Developing Children: An Experimental Study

**DOI:** 10.3390/ijerph17114071

**Published:** 2020-06-07

**Authors:** Martina Miklós, Dániel Komáromy, Judit Futó, Judit Balázs

**Affiliations:** 1Doctoral School of Psychology, Institute of Psychology, ELTE Eötvös Loránd University, Izabella St. 46, 1064 Budapest, Hungary; 2Department of Developmental and Clinical Child Psychology, Institute of Psychology, ELTE Eötvös Loránd University, Izabella St. 46, 1064 Budapest, Hungary; d.komaromy@student.vu.nl (D.K.); futo.judit@ppk.elte.hu (J.F.); balazs.judit@ppk.elte.hu (J.B.); 3Department of Behavioral and Movement Sciences, Vrije Universiteit, De Boelelaan 1105, 1081 HV Amsterdam, The Netherlands; 4Faculty of Social and Behavioural Sciences, University of Amsterdam, Nieuwe Achtergracht 166, 1018 WV Amsterdam, The Netherlands; 5Department of Psychology, Bjørknes University College, Lovisenberggata 13, 0456 Oslo, Norway

**Keywords:** physical activity, attention, executive function, attention-deficit hyperactivity disorder, ADHD, typically developing children

## Abstract

A growing number of studies support the theory that physical activity can effectively foster the cognitive function of children with attention-deficit hyperactivity disorder (ADHD). The present study examines the effect of acute moderate physical activity on the executive functions and attention performance of (1) typically developing children (without psychological, psychiatric or neurological diagnosis and/or associated treatment stated in their medical history); (2) treatment-naïve ADHD children; and (3) medicated children with ADHD. In the current study, a total sample of 150 (50 non-medicated, 50 medicated, and 50 typically developing) children between the ages of 6 and 12 took part in the experiment. The Mini International Neuropsychiatric Interview for Children and Adolescents (MINI Kid) was used to measure ADHD and the child version of the Test of Attentional Performance (KiTAP) was applied to evaluate the children’s attentional and executive function performance before and after two types of intervention. In order to compare the effects of physical activity and control intervention, half of the children from each group (25 participants) took part in a 20-min long, moderately intense physical activity session on the 60–80% of their maximum heart rate, while watching a cartoon video. In the control condition, the other half of the children (25 participants) from each group watched the same cartoon video for 20 min while seated. Physical activity (compared to the just video watching control condition) had a significantly positive influence on 2 out of 15 measured parameters (median reaction time in the alertness task and error rates in the divided attention task) for the medicated group and on 2 out of the 15 measured variables (number of total errors and errors when distractor was presented, both in the distractibility task) regarding the treatment-naïve group. Future studies should focus on finding the optimal type, intensity, and duration of physical activity that could be a potential complementary intervention in treating deficits regarding ADHD in children.

## 1. Introduction

Attention-deficit hyperactivity disorder (ADHD) is characterized by inattentive, impulsive, and hyperactive behavior [1], with a prevalence of 4–6% among school-age children [2,3]. ADHD is under the “Neurodevelopmental Disorders” section in the Diagnostic and Statistical Manual of Mental Disorders, 5th edition (DSM-5) [4]. In the case of ADHD, abnormalities were found in the fronto-cortical and fronto-striatal regions and systems of the brain [5].

The frontal region and especially the prefrontal cortex of the brain are involved in “executive function” (EF) tasks, which are higher-order cognitive tasks, basically including inhibition, working memory, and switching [6,7]. Some studies suggest that reasoning, problem-solving, and planning are based on these aforementioned processes [8,9]. Some researchers suggested that hyperactive, impulsive, and inattentive behaviors of children with ADHD are consequences of EF deficits (e.g., [10,11,12]).

Dysfunctions in the level of neurotransmitters are also apparent in ADHD [13]. Abnormal dopamine and norepinephrine levels were measured in the prefrontal cortex, the area of the brain responsible for certain symptoms of ADHD, such as deficits in inhibitory and executive control regarding attention [14]. Hence, sympathomimetic drugs such as methylphenidate (MTP), amphetamines or atomoxetine are often used for the treatment of ADHD by increasing the level of catecholamines (i.e., dopamine and norepinephrine) in the prefrontal cortex [14,15].

Several studies examined the effects of methylphenidate [16,17] and atomoxetine [18,19,20,21] for pharmacological treatment in ADHD. In addition to medication, the multimodal treatment of ADHD includes non-medicated treatments, such as cognitive-behavioral therapy [22], mindfulness-based exercise [23], social skills training [24], and parent training [25]. Up until now, the most effective treatment for reducing ADHD symptoms is a combination of medicated and non-medicated treatments [2,26,27].

Physical activity seems to influence at least some neurochemicals that are important in ADHD [28]. For example, brain-derived neurotrophic factor (BDNF) is associated with the dopaminergic neurons by executing their differentiation and survival [28]. A decreased level of BDNF characterizes individuals with ADHD [29], which might contribute to the dysfunctions of the dopaminergic system observable in this neurodevelopmental disorder [30]. Acute exercise is known to increase the levels of BDNF [31], and therefore might ameliorate the symptoms associated with ADHD.

Review articles and meta-analyses point to the beneficial effects of acute exercise on cognitive performance, particularly in children [32,33,34,35,36,37]. The literature also supports the beneficial effects of physical activity specifically on EFs [38,39,40].

Several review articles focus on the role and effectiveness of physical activity interventions on ADHD [41,42,43,44,45,46,47]. Further, numerous studies were conducted in the past decade to examine the effects of acute physical activity on EFs and attention, especially in children with ADHD [48,49,50,51,52,53,54,55,56]. There is a lack of consistency as to the type, duration, and intensity of physical activity applied. Furthermore, the examined EFs, test batteries, and experimental designs show great variability as well. Another source of difficulty is that the type of ADHD in the studies and the adjusted medication are often not controlled or not well controlled; moreover, the age and gender distribution of the participants also varies greatly between studies. Studies using acute physical activity had different conclusions: some of them displayed improvement in at least one executive function area [49,50,51,52,53,54,55,56]. Half of these eight studies showed improvement in inhibitory function [50,52,53,55], while others found amelioration of cognitive flexibility [55] and task switching [54] after 20 min (or overall 30 min with 20 min of main exercise) of acute physical activity.

In our current study, we aimed to fill the gap in the literature by examining the effects of a single bout of moderately intense exercise on attention and EFs with the same methodology in treatment-naïve and medicated children diagnosed with ADHD and in typically developing children. Our research question was about the effectiveness of acute physical activity compared to control intervention in all study groups based on the applied neuropsychological subtests. We hypothesized the following significant improvements from pre- to post-test measures after physical activity compared to the control condition in all study groups: (1) faster reaction time and less reaction time variability in alertness task; (2) less total omissions, omissions with distractor, omissions without distractor, number of total errors, errors with distractor, and errors without distractor in distractibility task; (3) faster reaction time, less numbers of total omissions, and total errors in divided attention task; (4) faster reaction time and less number of total errors in flexibility task; and (5) faster reaction time and less number of total errors in go/no-go task. An additional question was whether the expected differences between the two interventions would differ between the three groups.

Different research questions were already reported by our group from the same studies [57,58], using the same instruments and dataset, therefore similarities appear in the paper regarding the methodology and limitations.

## 2. Materials and Methods

### 2.1. Design and Participants

Three groups of children (typically developing children, treatment-naïve children with ADHD, and medicated children with ADHD) went through the following five stages in the experimental procedure: (1) informed consent; (2) diagnostic phase (structural diagnostic interview and demographic questionnaire); (3) pre-test (first neuropsychological test session); (4) interventions (physical activity or cartoon video as control); and (5) post-test (second neuropsychological test session). The design of the study is presented in Figure 1.

The whole experimental process took approximately 2.5 h per individual from the introduction of the study until finishing the last research phase.

Participants were recruited for the study between February 2016 and February 2018 [57,58]. A total of 168 children between 6–12 years were involved in the research. A final sample of 150 children remained following the exclusion of children who did not fulfill the definite inclusion criteria or met the exclusion criteria.

For the clinical sample (both treatment-naïve and medicated children with ADHD), participants were recruited to the Vadaskert Child Psychiatric Hospital and Outpatient Clinic, Budapest, Hungary [57,58]. Children in the treatment-naïve group were asking for professional help for the first time at the Vadaskert Child Psychiatric Hospital and Outpatient Clinic, Budapest, Hungary [57,58]. The diagnosis of ADHD was established here as well. The study examination processes had been conducted before the child’s psychiatrist suggested medication for the treatment of ADHD [57,58].

Regarding the control group, typically developing children were recruited from elementary schools in Budapest, Hungary [57,58].

The inclusion criteria for the clinical group were as follows: (a) age between 6–12 years; and (b) completed structured diagnostic interview (see below) establishing the diagnosis of ADHD [57,58]. Non-medicated children with ADHD and children undergoing adjusted medical treatment regarding their diagnoses and symptoms of ADHD created two clinical groups. The non-medicated sample was comprised of children with ADHD who had never been treated with any psychotropic medication for ADHD (treatment-naïve children with ADHD). The ongoing and adjusted medication for the medicated clinical sample was either methylphenidate or atomoxetine, which can be described in Hungary [57,58]. There were no significant differences between the physical activity and control intervention processes regarding the type of ongoing medication (χ^2^ (1) = 0.17, *p* > 0.05).

For the control group, the inclusion criteria stated that: (a) age must be between 6–12 years; (b) the participants do not have any psychological, psychiatric or neurological diagnosis and/or associated treatment stated in their medical history based on their parents; and (c) the absence of ADHD ratified by the structured diagnostic interview [58].

For all study groups the exclusion criteria were the following: (a) intellectual disability in the medical history; (b) autism spectrum disorder in the medical history; (c) oppositional behavior (because of low motivation) during the tests; (d) ongoing illness (e.g., diarrhea, stomach ache); (e) use of other psychotropic medications; and (f) an unfinished diagnostic interview; (g) retrospectively established autism or intellectual disabilities [57,58]. Furthermore, exclusion criteria in the control group were any former psychological, psychiatric or neurological treatment in medical history, based on their parents.

Children were excluded from the experimental (physical activity) session and the study if they: (a) had congenital or acquired heart disease; (b) another type of cardiovascular disease; (c) asthma; or (d) diabetes in their medical history. Furthermore, children could not participate in the acute physical activity session—and were therefore excluded from the study—(e) if their resting heart rate was above 110 bpm and/or their blood pressure was above 130/80 mm Hg at the beginning. The study was approved by the Ethical Committee of the Medical Research Council, Hungary (ETT-TUKEB-5677-1/2016/EKU [89/16]) [57,58]. All of the children and the parents of each child took part in this research after being informed about the nature of the study and subsequently providing a written informed consent.

### 2.2. Experimental Protocols and Procedures

The children’s distribution by interventions and groups is presented in Table 1.

From the 50 non-medicated, 50 medicated, and 50 control children, 25 carried out the physical activity intervention while watching the cartoon video in each group (non-medicated, medicated, and control), and the other 25 children participated in the control (only cartoon video watching) condition in each group, as well. Children from all study groups were randomly assigned to one of the intervention types.

Before this first KiTAP phase, children in the medicated group took their medication if it was methylphenidate 1–1.5 h before testing; in the case of treatment with atomoxetine, the effect of medication was continuous. After this, depending on whether the place of exercise was available or not, children participated in the exercise or the control condition.

Prior to the exercise condition, the resting heart rate and blood pressure were registered. If the measured parameters met the inclusion criterion, the minimal and maximal target zones for acute physical activity were calculated by taking 60% and 80% of the maximal heart rate (HR_max_ = 220 − age) [59,60]. This recommendation about intensity and exercise protocol was proposed by the head of the Performance Diagnostics Research Department of the National Sports Medical Institute (Sports Hospital) in Budapest, Hungary. Therefore, (220 − age) × 0.6 was used to get the minimal target zone and (220 − age) × 0.8 was used to get the maximal target zone for each child individually. The intensity of exercise was maintained between these target zones. Heart rate was measured during the whole workout with a heart rate monitor chest strap, placed on the child’s chest. The running activity was carried out in the form of interval training, in which the total time of 20 min was divided into 4 × 4 periods, with 1-min slow walking “breaks” between each period. During this physical activity phase, children watched a 20-min long cartoon video (Penguins of Madagascar, 2 parts, sum 20 min; the same as for control group children). A 5-min warm-up phase was conducted before the training, and 4 min of rest were allowed for the children before starting the second testing session with KiTAP.

In the control condition, children watched a cartoon video (Penguins of Madagascar, same as in the exercise condition) for 20 min while seated.

After each condition, the second round of neuropsychological tests were carried out. When finishing the experiment, each child received a personal certificate of accomplishment.

### 2.3. Measures

#### 2.3.1. Mini International Neuropsychiatric Interview for Children and Adolescents (MINI Kid)

During the experiment, firstly the modified version of the Hungarian Mini International Neuropsychiatric Interview for Children and Adolescents (MINI Kid) was used for diagnosing psychiatric disorders [61,62,63,64,65]. Applying MINI Kid either established (in case of treatment-naïve and medicated children with ADHD) or excluded (in case of control group children) the diagnosis of ADHD. MINI Kid is a short, structured diagnostic interview, which assesses 25 child psychiatric disorders according to the Diagnostic and Statistical Manual of Mental Disorders, 4th Edition (DSM-IV) [66]. The modified version also measures subthreshold disorders. The concurrent validity and reliability of the MINI Kid was examined by Sheehan and colleagues [65], and all the examined parameters gave acceptable results. Balázs (last author of the manuscript) and colleagues [64,67] developed the Hungarian version of the MINI Kid, which was applied in our study. Both inter-rater and test-retest reliability of the questionnaire was adequate for the analyzed psychiatric disorders and the criterion validity of the measure was also found acceptable when it was tested with sensitivity and specificity values as reported by their study [52]. Although the Hungarian version of the DSM-5 based version was already evaluated, at the beginning of the current study it was not accessible. In the diagnostic phase, parent–child dyads were interviewed, in accordance with the instructions of the MINI Kid administration process [64,65]. This experimental stage took approximately 45 min. The structural diagnostic interview was administered by the first author of this paper (M.M.), who is a psychologist. She finished MINI Kid training prior to starting the study and her activity was constantly supervised throughout the whole research process by another author of this study (J.B.), who is a child psychiatrist.

#### 2.3.2. Demographic Questionnaire

The demographic questionnaire was a structured parent-rated questionnaire, evolved specifically for our studies [57,58] to gather information about the demographic data of the participants, like parents’ education and economic activity, the structure of the family, as well as former and current psychological, psychiatric or neurological treatment.

#### 2.3.3. KiTAP

The KiTAP (Testbatterie zur Aufmerksamkeitsprüfung für Kinder) is a computer-based continuous performance task (CPT) and executive function battery [68,69]. It is the child version of the Tests of Attentional Performance (TAP), which was originally developed to examine attention and EF performance in adults with diverse medical, neurological, and psychiatric conditions. For the purpose of motivating young children to accomplish such tests, KiTAP was designed as child-friendly by creating the tasks as part of an enchanted castle story. Although the battery contains eight subtests, in this study we used only five of them: alertness, distractibility, divided attention, flexibility, and reaction control (inhibition, go/no-go) [68]. The rationale for this subtest choice was the shorter test duration of the aforementioned tasks in order to maintain children’s motivation to carry out the neuropsychological tasks in the post-testing phase as well.

A simple reaction test, called ‘The Witch’, was used to examine intrinsic alertness. In the task a witch appears at a window, and the aim is to make her disappear as fast as possible by pushing a reaction key button [68]. The distractibility subtest (named as the ‘Happy and Sad Ghosts’) is a centrally presented decision task. Half of the trials include a distracting stimulus, popping up in the periphery [68]. Either a happy or sad ghost appears as the central stimulus. The only visual difference between them is their mouth-line. Just one saccade is possible to the distracting stimulus before the central stimulus occurs, while the distractor emerges 400 ms earlier than the central stimulus. The central stimulus appears for 200 ms and typically disappears before fixation on it can happen. Omissions of the central stimulus or false reactions (errors) can occur because of the switch in the focus of attention caused by the distractor. A dual task (called ‘The Owl’) is used for assessing divided attention. The participants are asked to listen to a series of high and low tone owl sounds, as well as pay attention to the target stimuli, which is an owl with closed eyes [68]. When a sound is repeated or when the target stimulus shows up, the participants must reply by pressing the reaction button. The ‘The Dragon’s House’ task was created to examine the ability of flexibility. The purpose of the task is to vary the attentional focus between recognizing a blue and a green dragon popping up simultaneously on the two sides (left and right) of the screen. Attentional focus has to be varied between recognizing the different dragons by tapping one of two buttons (numbered as 1 and 2) [68]. The fifth subtest, called ‘The Bat’, is used for assessing inhibition (go/no-go task). Here the participants are required to react when the target stimulus (a bat) emerges in the middle of the screen by pushing the reaction key button, but they should not give response when the non-target stimulus (a cat) shows up [68]. The ability of control and decision-making are explored by expecting a fast reaction for one stimulus and still demanding no response for the other.

Approximately 30 min was required to complete these five KiTAP subtests. The subtests were carried out in a quasi-randomized order [58]. A research assistant supported the participants during this phase. At the beginning of the experimental phase, before starting each subtest, the research assistant described the goal of the test. The participant then completed a short pre-test at the first session of neuropsychological testing, allowing them to understand the task [58].

Figure 2 displays the experimental setup of KiTAP.

#### 2.3.4. Equipment Used in the Exercise and Control Condition

From all study groups, children’s blood pressure and resting heart rate were measured in seated position before taking part in physical activity. A TC7 Treadmill DOMYOS was used for the acute physical activity, while heart rate was measured continuously with a heart rate monitor chest strap (Polar H7 HR Sensor WearLink Bluetooth and application). During both (exercise and control) conditions, children watched Penguins of Madagascar on an iPad.

### 2.4. Statistical Analysis

Statistical analysis was performed using R (3.5.1 version, R Foundation for Statistical Computing, Vienna, Austria). First, outlier detection was implemented on the bivariate dataset (pre- and post-intervention). Mahalanobis distance (MD) was used, which, based on a Chi-squared distribution, measures the extent to which cases are multivariate outliers. Here, we applied the commonly used *p* < 0.001 criterion [70]. The existence of one or more multivariate outliers is demonstrated by a maximum MD larger than the critical Chi-squared value for df = k (the number of predictor variables in the model) at a critical alpha value of 0.001.

Second, we inspected the distribution of every dependent variable by plotting density functions for each group–intervention combination and estimated descriptive statistics. As for continuous variables, we applied a Tukey power transformation to reduce the likelihood of having non-normal residuals in our models.

Third, mixed-effect models were estimated to account for both the between-subject factors (group membership and type of intervention) and the within-subject variation. This latter was achieved by adding a random intercept for each participant to the models. We estimated generalized linear mixed-effect models (GLMMs) assuming Gaussian distribution for continuous variables and Poisson or negative binomial distribution for the count variables. Accordingly, the test statistics for the regression parameters were *F*-values for continuous and *χ*^2^ values for count variables. For significance testing we adopted the widely-used *p* < 0.05 value. After running the regression models, we checked their diagnostics. As for the Gaussian models, we checked normality violation and heteroscedasticity both by visual inspection and by Shapiro–Wilk and Levene’s tests when necessary. After estimating Poisson models, we tested for potential overdispersion and zero-inflation. If the overdispersion test was significant, or if the Poisson model did not converge, we estimated a negative binomial model. If there was still overdispersion in the model, we modeled it by groups (assuming that the overdispersion parameter is not identical for each participant but is a function of the group membership). If zero-inflation was present in the models, we modeled it first as identical for everyone, but if the first step did not lead to acceptable results, then it was modeled as a function of group membership. In the diagnostics, we tested for overdispersion and zero-inflation with the help of simulated residuals based on the model to compare empirical quantiles with the theoretical ones.

Fourth, we constructed effect plots for all main effects and interactions of the three independent variables (group, intervention, and time) with the Kenward–Roger degrees of freedom method. In this way we could visualize how each group performed on average before and after each intervention.

Finally, we computed estimated marginal means with Bonferroni adjustment to contrast time points. Contrasting the time points before and after the interventions (hence estimating the average change for each group–intervention combination) allowed us to compare the changes in time between the two types of interventions for the three groups. In these time contrasts, for easier interpretability, positive numbers indicate improvement from pre- to post-test. Hence, positive numbers mean a decrease in reaction time, number of omissions, number of errors, and other factors.

## 3. Results

### 3.1. Sample

After excluding those children who did not meet the definite inclusion criteria or met the exclusion criteria, finally the data of 50 treatment-naïve children with ADHD (45 boys and five girls, age: M = 8.26 years, SD = 1.47, aged 6–11 years), a further 50 children with ADHD receiving ongoing, adjusted medical treatment (47 boys and three girls, age: M = 9.70 years, SD = 1.78, aged 6–12 years) and another 50 children in the control group (43 boys and seven girls, age: M = 8.68 years, SD = 1.41, aged 6–11 years) were administered for this study. The full sample consists of mainly boys (135 boys versus 15 girls). There was a significant difference between the groups (F(2, 147) = 11.29, *p* < 0.001) regarding the ages. Table 2 includes the participants’ sociodemographic data.

In the non-medicated group, 40 children (80%) were diagnosed with the combined type, eight children (16%) with the mostly inattentive type, and two children (4%) with the mostly impulsive/hyperactive type of ADHD. The medicated sample consisted of 48 children (96%) with the combined type, one child (2%) with the mostly inattentive type, and one child (2%) with the mostly impulsive/hyperactive type of ADHD. As for the two clinical groups, diagnosis of ADHD was established by the structured diagnostic interview. Due to the small cases of mostly inattentive and mostly impulsive/hyperactive types, only the combined type was compared with Pearson’s Chi-squared test with Yates continuity correction. This combined type differed in accordance with group (χ^2^ (1) = 4.6, *p* = 0.03), indicating more combined diagnoses in the medicated group.

In total, 43 (86%) children received methylphenidate treatment, whereas seven (14%) used atomoxetine in the medicated clinical group. The average dose of methylphenidate was 15.7 mg (SD = 7.68), and the average dose of atomoxetine was 39.3 mg (SD = 15.66). On the day of testing all children from the medicated group took their prescribed adjusted medication.

The mean of the average heart rate during exercise was 140.57 bpm (SD = 5.92 bpm). There was no significant difference (F(2, 72) = 0.43, *p* > 0.05) between the groups regarding this parameter.

### 3.2. KiTAP Parameters

Significant baseline differences between the types of intervention were found only for errors in the go/no-go task (Wilcoxon W = 2141, *p* = 0.02). Table 3 presents all the relevant main effects and interactions.

Significant group effect indicates that groups performed differently across intervention types and time points, whereas significant time main effect implies significant changes from pre- to post-test among all groups and interventions.

#### 3.2.1. Alertness

##### Median of Reaction Time

Outlier detection found five outliers for median reaction time. As can be seen from Figure 3, the median of reaction time increases in each case, and hence the question was if the physical activity intervention decreased this increment compared to the control intervention.

A linear mixed-effect model with Gaussian distribution was used on the Tukey transformed data. The intervention and time interaction showed marginal significance (F(1, 139) = 3.51, *p* = 0.06), suggesting a somewhat different effect of the two interventions for all groups. The performance worsened after each condition in all groups, and the effect of the interventions did not differ significantly as for the non-medicated and control groups (see Figure 3 above).

Conversely, the biggest difference (smallest overlap between confidence intervals) between the types of interventions (physical activity vs. control) was observable for the medicated group, resulting in significant difference between the two types of intervention in the post-hoc testing. Compared to the physical activity intervention (95% confidence interval (CI): −0.0039; −0.0005), reaction time increased significantly more (t(139)= 2.369, *p* = 0.02) in the control condition (95% CI: −0.0067; −0.0033). For the control group, the two interventions had practically the same effect, while for the non-medicated children, running increased the reaction time somewhat less (see Figure 4).

##### Variability of Reaction Time

As for the standard deviation (variability) of reaction time, five outliers were detected. The means and standard deviations of variability of reaction time increased from pre- to post-test for all groups in both interventions. The effect of intervention did not differ significantly for any of the groups.

#### 3.2.2. Distractibility

The number of omissions (total, with and without distractor) was quite low in the pre- and post-tests and changes from pre- to post-test were not considerable.

##### Total Omissions

Four outliers were detected for total omissions. None of the groups exhibited significantly different changes over time between the two interventions, and the smallest overlap between confidence intervals was detected among the medicated children.

##### Omissions with Distractor

Six outliers appeared for omissions with distractor. There were no relevant differences between the two interventions either at the regression analysis or at the post-hoc tests.

##### Omissions without Distractor

Four outliers were detected for omissions without distractor. Neither the regression analysis nor the post-hoc tests displayed any significant differences between the two types of interventions.

##### Total Error

Descriptive data show that the number of errors decreased from the first time point of testing to the second for both types of intervention (Figure 5). The question is whether this decline is greater for acute physical activity than for the control condition.

Negative binomial regression showed significant group and time interaction (χ^2^ (2) = 8.87, *p* = 0.01), possibly due to the higher performance increment of medicated and control groups compared to the non-medicated group. Consequently, the non-medicated group performed worse than the other two groups (see Figure 5 above).

The difference between the two interventions was significant (z = 2.24, *p* = 0.03) for the non-medicated group. Whereas the control intervention had only a mild effect (95% CI: −0.0573; 0.191) and did not significantly differ from zero, the exercise condition yielded significantly greater improvement (95% CI: 0.1414; 0.4826) (see Figure 6).

##### Errors with Distractor

Next, we assessed the number of errors when distractors were presented in the task. For all groups in both conditions, there was a decrease over time in the number of errors (Figure 7).

Poisson GLMMs with added zero-inflation parameters displayed group and time interaction as marginally significant (χ^2^ (2) = 5.97, *p* = 0.05), showing some difference among groups in performance-change over time. A three-way interaction between intervention, time, and group was also marginally significant (χ^2^ (2) = 5.18, *p* = 0.08), suggesting that the two interventions exerted a different impact on the different groups over time (see Figure 7 above).

Similar to the previous measure, the effects of the two interventions differed significantly for the treatment-naïve group. Exercise intervention (95% CI: 0.2257; 0.6569) decreased the error rates significantly more (t(286) = 2.682, *p* = 0.008) than the control intervention, which had no significant effect (95% CI: −0.1236; 0.2388). This result implies significantly greater improvement in the post-test for the non-medicated group after acute physical activity (see Figure 8).

##### Errors without Distractor

Regarding the number of errors without distractor, all groups in both conditions showed improvement over time. Group and time interaction was significant (χ^2^ (2) = 11.33, *p* = 0.004) by Poisson GLMM regression. This might be explained by the smaller decrease in errors for the treatment-naïve group than for the other two groups. Post-hoc tests did not show any significant differences between the two types of interventions.

#### 3.2.3. Divided Attention

##### Median of Reaction Time

A decrease in the median reaction time was found for all groups in both interventions. Neither the regression analysis nor the post-hoc tests presented any significant differences between the two types of interventions.

##### Total Omissions

Four outliers were found as to the number of omissions. In the medicated group, both interventions did not result in virtually any change. In the control group, the control condition slightly increased the average number of omissions, whereas after running, the average number remained unchanged. Finally, among treatment-naïve children, the average number of omissions increased after the control intervention and dropped after exercise (Figure 9).

A marginally significant time main effect (χ^2^ (1) = 3.63, *p* = 0.06) as well as a significant intervention and time interaction (χ^2^ (1) = 8.91, *p* = 0.003) were found by Poisson GLMM, showing that the changes in performance over time differed between interventions. It plausibly reflects the fact that compared to the control intervention, the exercise condition decreased the number of omissions, and hence, improved performance (see Figure 9 above).

There is a significant difference (z = 2.818, *p* = 0.005) for the treatment-naïve group between the control (95% CI: −0.4866; −0.036) and exercise (95% CI: −0.0269; 0.4879) interventions. Whilst the control intervention significantly increased the number of omissions, children in the exercise condition accomplished the same performance in the two time points (see Figure 10).

##### Total Error

Six outliers appeared for number of errors. Descriptive data showed very slight improvement for control children in both interventions (Figure 11).

The Poisson GLMM detected a significant group and time interaction (χ^2^ (2) = 10.07, *p* = 0.007), and a significant three-way interaction between group, intervention, and time (χ^2^ (2) = 9, *p* = 0.01). It implies that the patterns in performance changes significantly differ between groups (probably due to the greater improvement in the treatment-naïve group), as well as the effect of the exercise (compared to the control condition), which did not have the same impact on all groups between the two time points (see Figure 11 above).

Accordingly, the post-hoc tests show that for the medicated group, while the exercise intervention significantly improved the performance (95% CI: 0.3095; 0.7331), the control intervention did not significantly change the number of errors (95% CI: −0.1329; 0.2177), resulting in a significant contrast between the two interventions (z = 3.459, *p* < 0.001) (see Figure 12). Although the treatment-naïve group exhibited great improvement, the effect of the exercise, however, did not differ from that of the control condition.

#### 3.2.4. Flexibility

##### Median of Reaction Time

One outlier was detected regarding median reaction time. A decrease in means was detected for all groups in all interventions. The group and intervention interaction exhibited marginal significance (F(2, 143) = 2.93, *p* = 0.06) by linear mixed-effect model estimation on Tukey transformed data. The three groups displayed very similar patterns over time, and the effects did not differ between the two interventions. Neither the regression analysis nor the post-hoc tests showed any significant results.

##### Total Error

Two outliers were found regarding the number of errors. Means decreased from pre- to post-test and all groups exhibited a very similar performance pattern over time. The effects did not differ markedly between the two types of interventions. Neither the regression analysis nor the post-hoc tests showed any significant results.

#### 3.2.5. Go/No-Go

##### Median of Reaction Time

No outliers were detected regarding the median reaction time. After the control intervention, the performances either worsened or stagnated, and the exercise condition somewhat reduced the reaction time for the control group and increased it in the other two groups. The two-way interaction between these variables (group and time) also reached significance (F(2, 144) = 3.26, *p* = 0.04), and the intervention and time interaction proved to be marginally significant (F(1, 144) = 3.86, *p* = 0.05). On the one hand, these results suggest that different groups exhibited different time trends, independently of intervention conditions. On the other hand, they suggest that the two types of interventions had different impacts over time on the performance (independently of group membership), albeit only marginally significantly. Among control children, the running intervention decreased reaction time, whereas the control intervention essentially did not change it. Similarly, for the medicated children, the control intervention deteriorated task performance much more than the exercise intervention. The effects of the two types of intervention were considerably similar to the treatment-naïve group. Time contrasts between interventions did not reach significance for any groups.

##### Total Error

Two outliers emerged concerning the number of errors. Baseline differences were significant for interventions (Wilcoxon W = 2141, *p* = 0.02, exercise condition: M = 2.173, control condition: M = 3.260), suggesting that participants of the two intervention conditions differed in performance before assignment. Intervention and time had a marginally significant interaction (χ^2^ (1) = 3.6, *p* = 0.06) resulting from a negative binomial GLMM. Interestingly, this result shows that there is a tendency for greater change in performance (and fewer errors) after the control intervention in the post-test. Time contrasts had the smallest overlap for control children, resulting in a marginally significant (z = −1.937, *p* = 0.053) difference between the two types of intervention. This implies a tendency for a decreased number of errors in the control condition and no relevant effect for the exercise condition for control children.

## 4. Discussions

To our knowledge, this is the first study to examine the effect of acute physical activity on various aspects of attention and executive functions in treatment-naïve and medicated children with ADHD and in typically developing children in one research setting. In general, children with the most severe symptoms (having the combined diagnosis of ADHD) were on medication, while the “less severe” cases (either inattentive or hyperactive types) were not. This ascertainment might be the reason for the significantly greater number of combined type ADHD regarding the clinical groups.

The main effect of groups was significant in 14 out of 15 parameters across time points and interventions, implying significantly worse performance for the non-medicated group.

Regarding the alertness task, the median of reaction time increased significantly after both interventions in all groups. Although the GLMM regression analysis did not display significant interactions, post-hoc testing found a significant difference between the two types of interventions as for the medicated group, resulting in a greater increment in reaction time after the control intervention. Furthermore, compared to the pre-tests, a significant increment was found in the variability (standard deviation) of reaction times in the post-tests for all groups, without significant interactions in the GLMM regression analysis. The opposite result was found in other studies, suggesting positive effects in speed processing both in children with and without ADHD after acute physical activity with moderate intensity [53], and also significantly faster response speed and lower response speed variability after high intensity treadmill exercise session in boys with ADHD [49]. In contrast with the aforementioned results [49,53], a significant decrease was found in the median of reaction time in the divided attention and flexibility tasks for all three groups after the acute physical activity intervention, but interestingly, a similar decrement was found following the control condition, too. As for these tasks, there was no significant difference between the effects of the two interventions regarding any of the groups between the two time points. The significant two-way interaction between group and time in the go/no-go task suggests that regardless of the intervention type, the performance of the groups differed over time. In this task, among control children, physical activity decreased reaction time, while the control intervention essentially did not change it. For medicated children, the control condition worsened task performance more than the exercise intervention (both interventions increased the reaction time), albeit without significance. The impacts of the two types of intervention were considerably similar to each other in the treatment-naïve group. In contrast, one study [71] using exergaming exercise found improvement as to reaction time in inhibition and switching.

In the distractibility task, a very low number of omissions were apparent in the pre- and post-tests, and consequently, changes from pre- to post-test were not substantial. A low degree of distractibility is critical for academic achievement and work [68]. There were no significant interactions regarding the GLMM regression analysis for the number of total omissions and omissions either with or without distractors. The differences between the two types of interventions were also not significant by these parameters. The marginally significant intervention main effect for the omissions made with the distractor suggests that the two types of interventions might bring about different performance change across groups and time points.

The elevated level of distractibility is a key criterion of ADHD in the DSM-5 [4]. This ascertainment was confirmed by our results by the significant main group effect in all of the three omission variables (total, with and without distractor). Contrary to our results, Van Mourik and colleagues [72] found that the presence of distractors could temporarily facilitate the performance of children with ADHD. Concerning the total number of errors in this task, the results implied that children in the medicated and control groups performed significantly better (making fewer errors) than the treatment-naïve group in the post-test, independently of interventions. Furthermore, in the non-medicated group, the significant difference between the exercise and control condition suggests that the physical activity significantly decreased the number of errors, whereas the control condition did not change it. As for the error rates in the distractor condition, both the group and time and a three-way interaction between group, time, and intervention were marginally significant. These results suggest that irrespective of interventions, the temporal performance of the groups might be different and that the two interventions might have had different impacts on the different groups over time. Additionally, a significant difference arose between the interventions regarding the non-medicated group, indicating that after physical activity, the number of errors made with distractor decreased significantly, whereas the control intervention had no effect. The reason behind the significant group and time interaction, concerning the errors made without distractor, may be that the non-medicated group produced a significantly smaller decrease than the other two groups. Additionally, for this parameter, all groups made significantly fewer errors in the post-test in both conditions than in the pre-test. Our finding that performance of the treatment-naïve group was considerably worse (more total errors, and errors with and without distractor), than that of the other two groups, has been supported by several studies [73,74,75,76]. Nevertheless, the positive effect of the acute physical activity was found in the non-medicated group, regarding the total number of errors and the errors made with distractor.

In everyday life, divided attention is necessary, since paying attention to several events and things at the same time occurs every day [68]. The subsequent ADHD criteria are related to impaired divided attention functioning in the DSM-5 [4]. Concerning our results as to omissions in the divided attention task, the significant interaction between intervention and time suggests that after the control condition, the performance worsened (increment in the number of omissions) compared to the exercise intervention, after which the performance improved. Regarding the non-medicated group, a significant difference was displayed between the interventions: whilst children in the control intervention made significantly more omissions, the members implementing the exercise condition accomplished the same performance in both time points. Significant two-way (group and time) and three-way (group, intervention, and time) interactions emerged for the number of errors, which might denote the greater improvement for the treatment-naïve group and also the effect of the exercise (compared to the control condition), which altered the performance differently between the groups between the two time points. Consequently, the post-hoc test showed that after physical activity, performance significantly improved, while the control intervention did not change performance significantly in the medicated group, producing a significant contrast between the two interventions. Contrary to our results, Elosúa and colleagues [77] exhibited fewer impairments by children with ADHD when compared to typically developing children in a divided attention task. Possible explanations for this contrast with our results might be the different age range (between 9–10 years versus 6–12 years) and the chosen digit recall task and paper-pencil based box-crossing task used for assessing divided attention. In addition, a study conducted by Lajoie and colleagues [78] found few differences between on-off medication as to sustained and selective attention parameters and processing speed, in contrast with measures of divided attention. In the latter task, greater accuracy was displayed with slower completion duration, resulting in a speed–accuracy trade-off, when children were on medication [78]. The reason for the discrepancy with our results might be due to the small sample size (*n* = 15) of their study [78] indicating limited statistical power to determine contrast between medication conditions. All children were on medication for at least eight months prior to the start of the experiment and also took a placebo as part of the off-medication condition. Meanwhile, treatment-naïve ADHD children in our study had never been treated with any medication. Another reason could be the different test battery applied by the researchers [78].

Various studies support the notion that impairment in cognitive flexibility/task switching is a characteristic of ADHD [79,80,81]. Our results are in line with this finding by the significant group main effect in the number of errors. However, according to the descriptive data, means decreased from pre- to post-test in all groups and all conditions, the three groups exhibited a very similar performance pattern over time, and the effects did not differ substantially between the two types of interventions. Accordingly, neither the GLMM regression analysis nor the post-hoc tests showed any significant results. These findings are contrary to those of Ludyga and colleagues [56] and Hung and colleagues [54], who indicated that both children with ADHD and typically developing children showed higher cognitive flexibility [56] following aerobic exercise, compared with the control intervention. Regarding the results of Ludyga and colleagues [56], the contrast with our results might be due to the participants’ older age range (between 11–16 years), the difference in experimental design, equipment (only post-test measures after the interventions; all participants took part in both interventions (physical activity and control); cycle ergometer), and the applied verbal cognitive flexibility task. In addition, regarding the study of Hung and colleagues [54], children with ADHD performed smaller global switching costs in reaction time after acute physical activity compared with following control sessions. This latter result is also the opposite of ours regarding the median of reaction time measured in the flexibility task: no significant difference was present between the two interventions; however, after both, the mean of the median reaction time decreased. The same possibilities could be mentioned as for the discrepancy between the results of Hung and colleagues [54] and ours: difference in the means of age, disparate experimental design (solely post-testing after taking part in both interventions), and different test (assessing task switching).

Deficits in inhibitory control have been already shown in individuals with ADHD in several studies [82,83,84,85,86,87]. A marginally significant intervention and time interaction might surprisingly show a tendency for greater change in performance (making fewer errors) after the control intervention in the post-test across all groups. Additionally, a marginally significant difference arose between the two types of intervention in the control group. This finding might display a tendency for a decreased number of errors in the control condition and no relevant effect for the exercise condition for control children. These results are opposite to those found in several other studies [50,52,53,55], indicating the positive effects on inhibition after 20 min (or overall, 30 min with 20 min of main exercise) of acute physical activity. This difference between the results might be the reason for using different tasks measuring inhibition (for example, the face validity of Stroop test as a method for assessing inhibition from interference [50]); completing only post-test measures [52,53]; taking part in physical activity as well as in control condition [52,53,55]; applying reading as control intervention [52]; carrying out the physical activity intervention on cycle ergometer [53,55]; and recruiting with older age range (between 11–16 years) [55].

The results of this study should be interpreted alongside the limitations, which mainly are drawn from the study design; this is why they are mostly the same as those reported by Miklós and colleagues [58]. First, significant age differences were found between the different groups, therefore it could have been a possible confounding factor in our study. Second, we were not able to examine control group children in the morning, because they had to attend school at this time. However, medicated children with ADHD had to be examined in the morning, because they had to take their methylphenidate medication in the morning and the examination had to be carried out while the children were under the effect of the medication (i.e., methylphenidate). Furthermore, regarding both treatment-naïve and medicated children with ADHD, we could not interrupt their therapeutic program they took part in at Vadaskert Child Psychiatric Hospital and Outpatient Clinic over that week. Therefore, it is possible that control children were more tired than clinical children throughout the phases of the experiment. Third, children were excluded from the control group if they had the diagnoses of autism spectrum disorder and intellectual disability and/or psychological, psychiatric or neurological treatment in their medical history, or if the structural diagnostic interview confirmed the diagnosis of ADHD. Nonetheless, children from any study group who met the criteria or subthreshold criteria for any other psychiatric disorder after accomplishing the MINI Kid were not excluded from the experiment process. Fourth, we did not assess the level of intelligence; however, children from all study groups were excluded if intellectual disability was stated in their medical history. Fifth, symptom severity could not be assessed without medication concerning the medicated group—whose members were receiving their prescribed dose of medication—because it would have raised ethical questions regarding withdrawing atomoxetine treatment just to conduct the study. Sixth, significant diagnostic differences were found between the medicated and non-medicated children regarding the combined ADHD subtype.

Regarding the internal validity, it is important to consider whether the observed changes in performance are due to the effects of the intervention or to the aforementioned limitations (i.e., age difference among the groups, differences in the time of the day the intervention was conducted, subtype of ADHD, symptom severity).

The aim of our study was to gather information that could form the basis of practical recommendations and interventions for children with ADHD, particularly within school settings in order to help manage their difficulties in attention and executive functions.

External validity of the study is high if interventions similar to our experimental design are implemented; for example, in the case of programs that encourage children with ADHD to engage in physical activity (specifically running at moderate intensity for at least 20 min) between classes in school. The external validity of the study is lower when long-term physical activity is concerned.

We took a step forward in verifying the reality of common knowledge about the advantageous effects of physical activity on children with ADHD. According to the results of our study, favorable effects of physical activity were relatively affirmed in the group of participating children with ADHD. Despite selective impacts on attention and executive function performances, the outcomes are promising for further studies. Future experiments are necessary to prove the effectiveness of physical activity on ADHD. Additionally, it is a relevant practical observation that even treatment-naïve children with ADHD could stay determined throughout a long experiment process in a personal situation (with the examiner only).

## 5. Conclusions

In summary, we would like to highlight that 20 min of moderate intensity physical activity had a positive and significantly different impact on two (median reaction time in the alertness task, error rates in the divided attention task) out of 15 parameters in the medicated group. A positive effect was measured also on two out of the 15 measured parameters (number of total errors and errors when distractor was presented in the distractibility task) for the treatment-naïve group. For the number of omissions in the divided attention task, performance did not change in the non-medicated group after physical activity, whereas the control condition significantly increased the omission rates. Our results partly supported the hypothesis that acute physical activity might have beneficial effects on attention and executive function performance, while strongly significant differences were not found by every parameter. Future studies should focus on finding the optimal type, intensity, and duration of physical activity that possibly could be a complementary intervention in treating deficits regarding ADHD in children.

## Figures and Tables

**Figure 1 ijerph-17-04071-f001:**
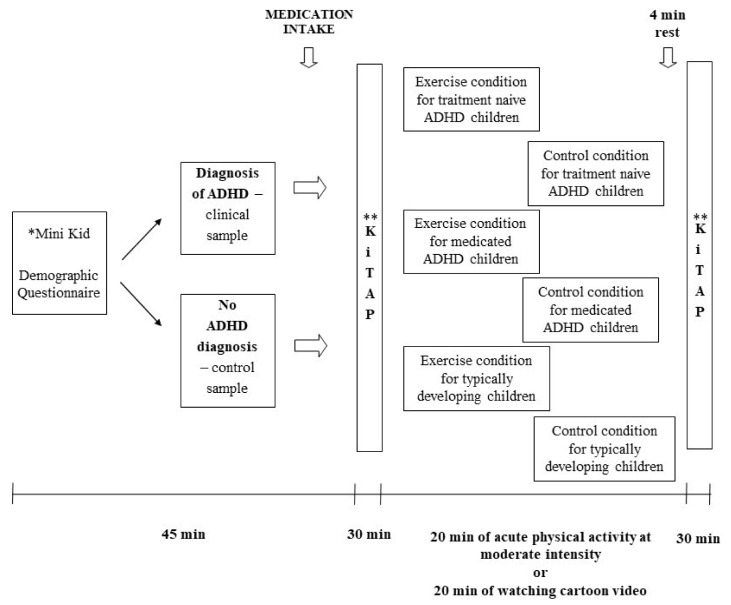
Study design. * MINI Kid: Mini International Neuropsychiatric Interview for Children and Adolescents; ** KiTAP: child version of the Test of Attentional Performance (or Testbatterie zur Aufmerksamkeitsprüfung für Kinder).

**Figure 2 ijerph-17-04071-f002:**
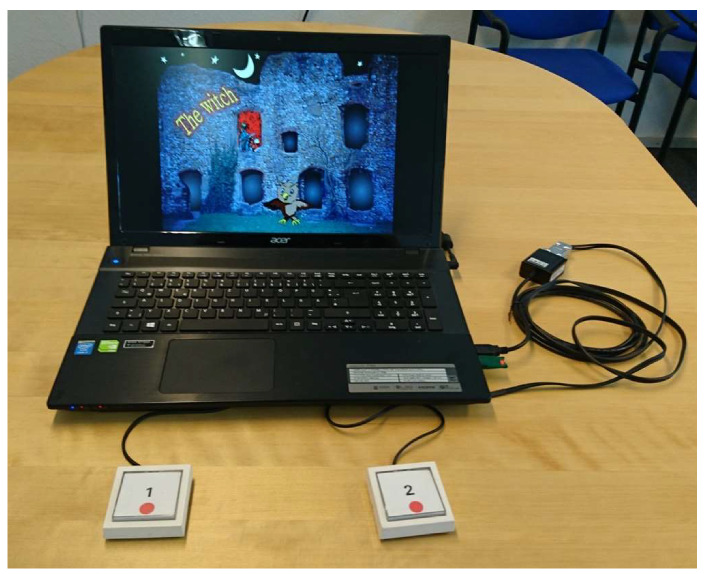
Experimental setup (promo picture from Psytest).

**Figure 3 ijerph-17-04071-f003:**
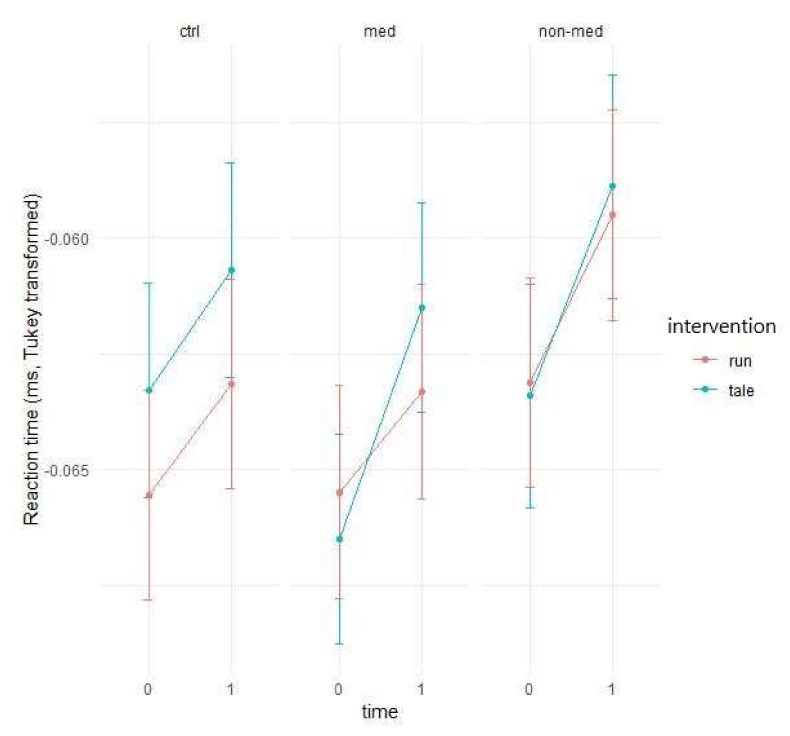
Prediction of the linear mixed model and alertness median of reaction time (Tukey transformed data).

**Figure 4 ijerph-17-04071-f004:**
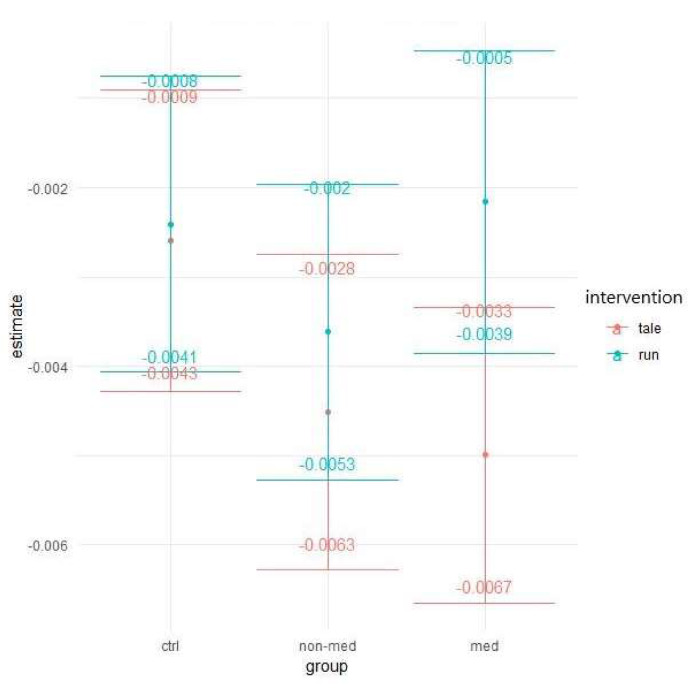
Time contrasts for alertness median of reaction time (Bonferroni correction).

**Figure 5 ijerph-17-04071-f005:**
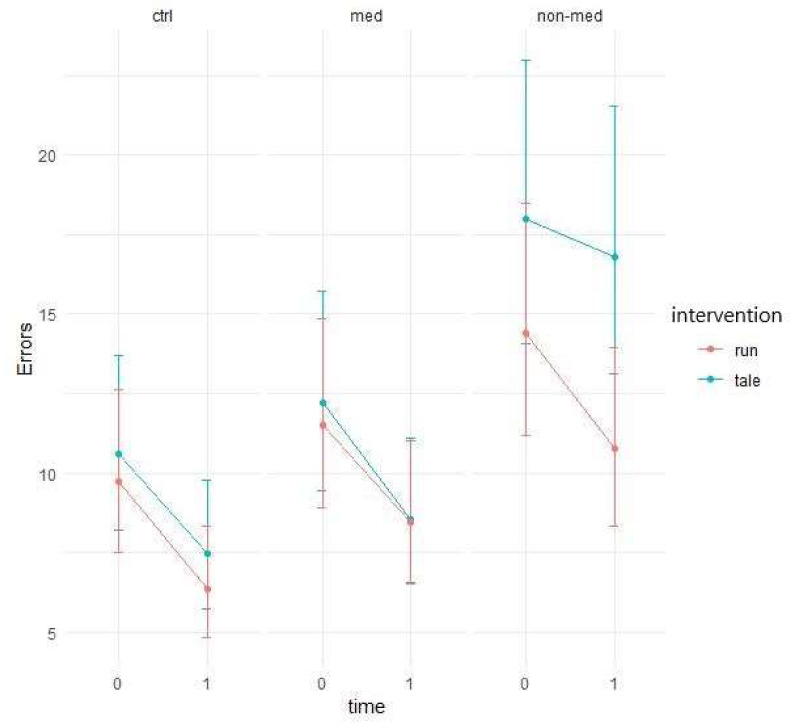
Prediction of the linear mixed model and distractibility total error (negative binomial regression).

**Figure 6 ijerph-17-04071-f006:**
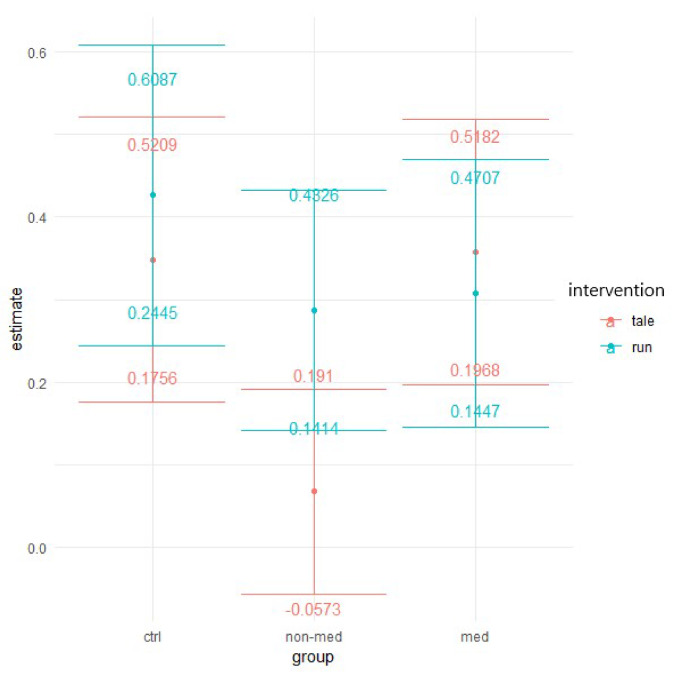
Time contrasts for distractibility total error (Bonferroni correction).

**Figure 7 ijerph-17-04071-f007:**
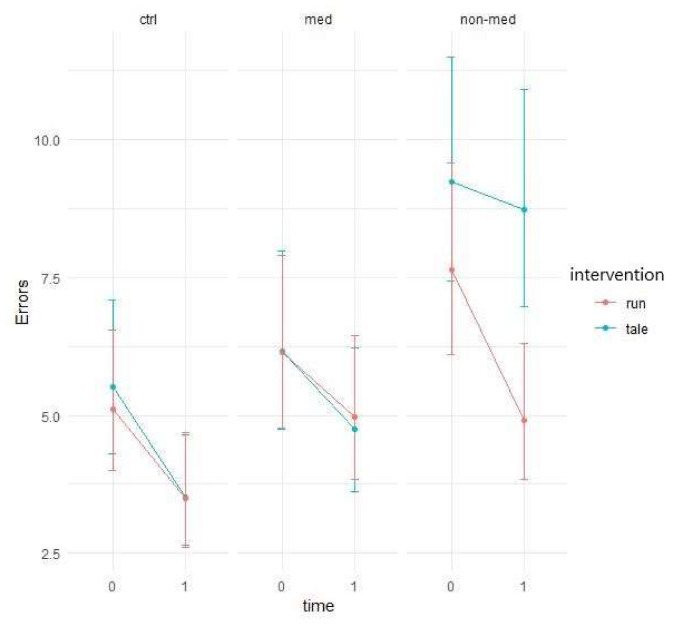
Prediction of the linear mixed model and distractibility error with distractor (Poisson regression with zero-inflation).

**Figure 8 ijerph-17-04071-f008:**
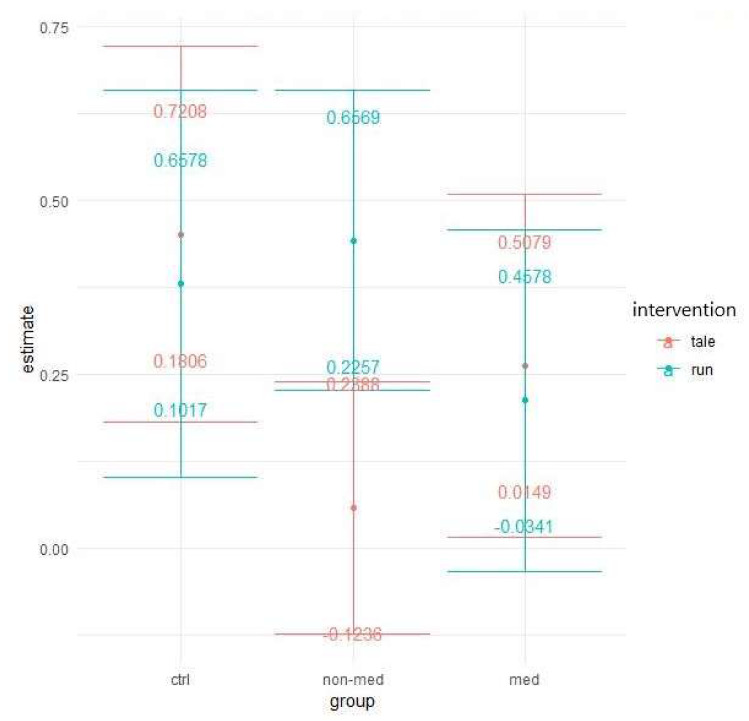
Time contrasts for distractibility error with distractor (Bonferroni correction).

**Figure 9 ijerph-17-04071-f009:**
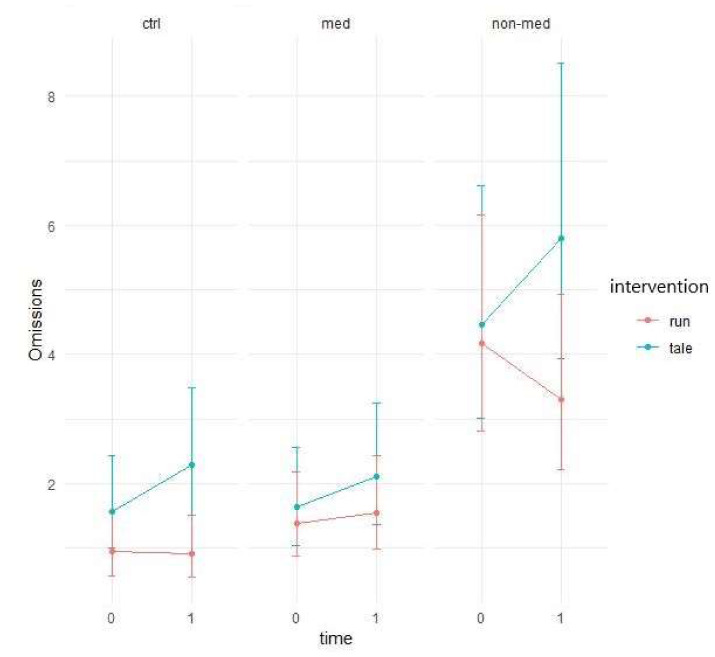
Prediction of the linear mixed model and divided attention total omission (Poisson regression).

**Figure 10 ijerph-17-04071-f010:**
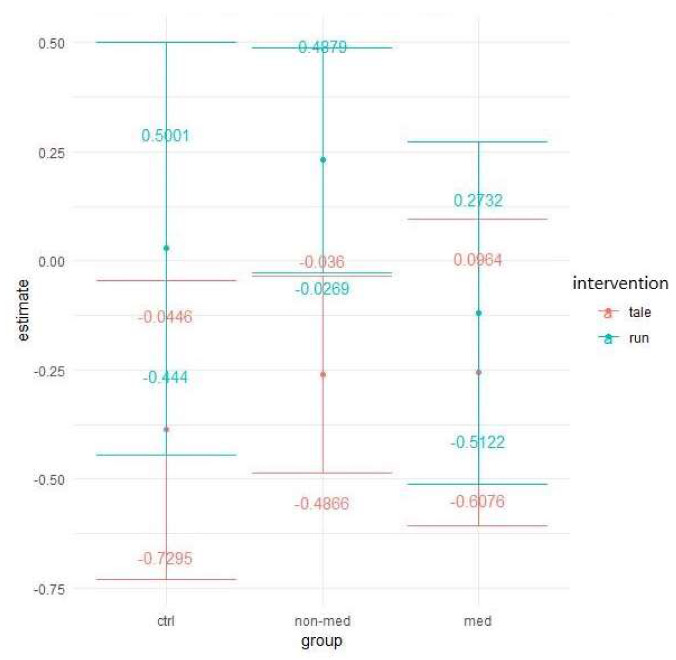
Time contrasts for divided attention total omission (Bonferroni correction).

**Figure 11 ijerph-17-04071-f011:**
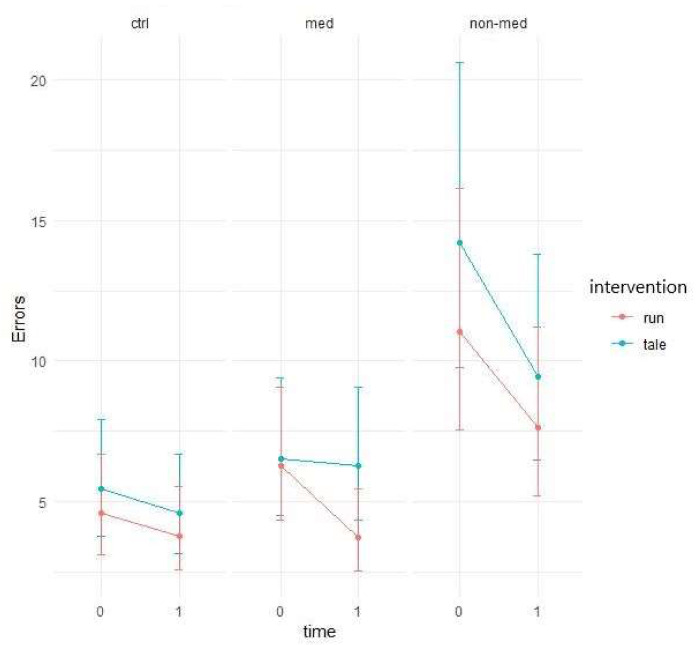
Prediction of the linear mixed model and divided attention total error (Poisson regression).

**Figure 12 ijerph-17-04071-f012:**
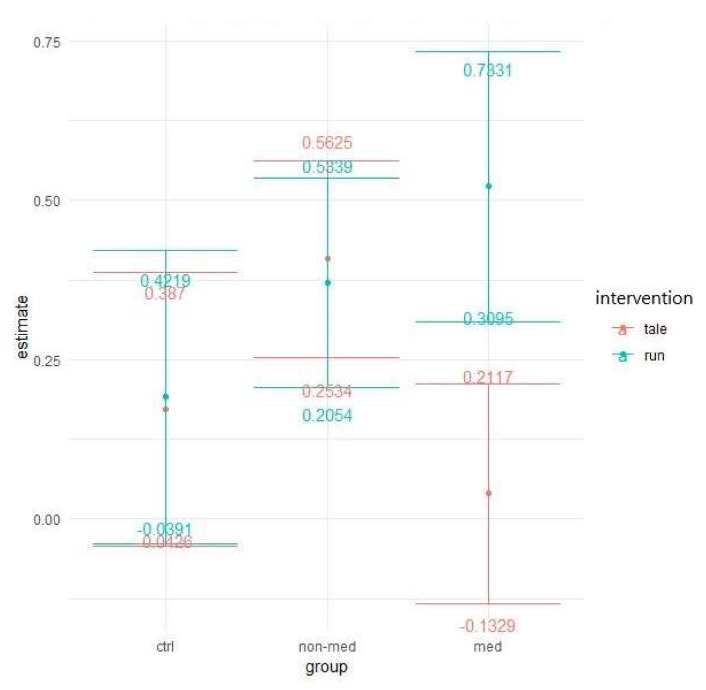
Time contrasts for divided attention total error (Bonferroni correction).

**Table 1 ijerph-17-04071-t001:** The children’s distribution by interventions and groups.

Groups Conditions	Non-Medicated Group (Number of Children)	Medicated Groups (Number of Children)	Control Group (Number of Children)
**Exercise**	25	25	25
**Control**	25	25	25
**Sum**	50	50	50

**Table 2 ijerph-17-04071-t002:** Sociodemographic data of the sample *.

Variables	Non-Medicated Group *n* = 50	Medicated Group *n* = 50	Control Group *n* = 50	*F* or Fischer’s Exact Test Value	*p*-Value
Age; mean (SD)	8.26 (1.47)	9.7 (1.78)	8.68 (1.41)	F(2, 147) = 11.29	***p* < 0.001**
Gender; persons (%)	45 boys (90%) 5 girls (10%)	47 boys (94%) 3 girls (6%)	43 boys (86%) 7 girls (14%)	Fischer’s exact test = 1.75	*p* > 0.05 (2-sided)
Residence; person (%)				Fischer’s exact test = 38.16	***p* < 0.001** (2-sided)
Capital	27 (54%)	21 (42%)	45 (90%)		
Countryside city	11 (22%)	23 (46%)	3 (6%)		
Village	12 (24%)	6 (12%)	1 (2%)		
Countryside town	0 (0%)	0 (0%)	1 (2%)		
Accommodation; person (%)				Fischer’s exact test = 5.23	*p* > 0.05 (2-sided)
Own parents	46 (92%)	47 (94%)	50 (100%)		
Adopted	3 (6%)	3 (6%)	0 (0%)		
Foster parents	1 (2%)	0 (0%)	0 (0%)		

* Source: demographic questionnaire. Significant *p* values (under 0.05) are shown in bold.

**Table 3 ijerph-17-04071-t003:** Main effects and interactions of each KiTAP variables

KiTAP Subtests	Main Effects and Interactions	Df	*F/**χ*^2^ Value	*p*-Value
**Alertness**	**Median of reaction time**			
	group	2	3.76	***p* < 0.05**
	time	1	94.29	***p* < 0.001**
	intervention: time	1	3.51	*p* > 0.05 (marginally)
	**Variability of reaction time**			
	group	2	14.17	***p* < 0.001**
	time	1	84.42	***p* < 0.001**
**Distractibility**	**Total omissions**			
	group	2	20.03	***p* < 0.001**
	**Omissions with distractor**			
	group	2	15.17	***p* < 0.001**
	intervention	1	3.66	*p* > 0.05 (marginally)
	**Omissions without distractor**			
	group	2	16.88	***p* < 0.001**
	Total error			
	group	2	20.86	***p* < 0.001**
	time	1	69.31	***p* < 0.001**
	group: time	2	8.87	***p* < 0.05**
	**Errors with distractor**			
	group	2	10.64	***p* < 0.01**
	time	1	10.79	***p* < 0.01**
	group: time	2	5.97	*p* > 0.05 (marginally)
	group: intervention: time	2	5.18	*p* > 0.05 (marginally)
	**Errors without distractor**			
	group	2	17.85	***p* < 0.001**
	time	1	33.62	***p* < 0.001**
	group: time	2	11.33	***p* < 0.01**
**Divided attention**	**Median of reaction time**			
	group	2	12.8	***p* < 0.001**
	time	1	28.88	***p* < 0.001**
	**Total omissions**			
	group	2	40.02	***p* < 0.001**
	intervention	1	6.34	***p* < 0.05**
	time	1	3.63	*p* > 0.05 (marginally)
	intervention: time	1	8.91	***p* < 0.01**
	**Total error**			
	group	2	14.37	***p* < 0.001**
	group: time	2	10.07	***p* < 0.01**
	group: intervention: time	2	9	***p* < 0.05**
**Flexibility**	**Median of reaction time**			
	group	2	4.56	***p* < 0.05**
	time	1	89.32	***p* < 0.001**
	group: intervention	2	2.93	*p* > 0.05 (marginally)
	**Total error**			
	group	2	8.62	***p* < 0.05**
	time	1	16.5	***p* < 0.001**
**Go/no-go**	**Median of reaction time**			
	group	2	2.73	*p* > 0.05 (marginally)
	time	1	6.06	***p* < 0.05**
	group: time	2	3.26	***p* < 0.05**
	intervention: time	1	3.86	*p* > 0.05 (marginally)
	**Total error**			
	group	2	25.49	***p* < 0.001**
	intervention	1	4.98	***p* < 0.05**
	time	1	3.99	***p* < 0.05**
	intervention: time	1	3.6	*p* > 0.05 (marginally)

Significant *p* values (under 0.05) are shown in bold.

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
