# Peer review of "Acute Physical Activity, Executive Function, and Attention Performance in Children with Attention-Deficit Hyperactivity Disorder and Typically Developing Children: An Experimental Study"

_ijerph, 2020, doi:10.3390/ijerph17114071_

Round 1

Reviewer 1 Report

Manuscript ID: ijerph-806811

Title: Acute physical activity, executive function and attention performance in children with attention-deficit hyperactivity disorder and typically developing children

Journal: International Journal of Environmental Research and Public Health

Abstract

1) Line 22. Authors should replace “treatment-naïve children” with “treatment-naïve ADHD children”.

2) Line 25. Authors should explain why just 25 children out of 50, from each group, participated in the physical activity session.

3) Lines 27-30. Authors should specify which parameters significantly improved and which task was used.

Introduction

1) Line 59. I believe that “neurochemicals” should be followed by a noun.

2) Line 76. Authors should replace “subjects” with “patients” or “participants” throughout the manuscript.

3) Line 84. Authors should put forward some research questions referred to the present manuscript as well as their hypotheses.

Materials and Methods

1) Lines 105-106. How did authors verify the b) criterion?

2) Lines 112-113. How did authors verify these criteria?

3) Lines 119-120. Authors should add the ethical committee report number.

4) Lines 147-149. Which was the rationale of this choice?

Results

1) Lines 264-268. Which is the meaning of “Data of treatment-naïve children with ADHD… was administered for this study”?

2) Line 269. Aiming to assess significant differences in age between groups, authors could perform an ANOVA.

3) Line 272. In Table 2 authors should add the results of statistical analyses referred to each variable.

4) Figure 2. The resolution of figures is quite low.

5) Lines 333-336. Authors should describe these significant effects.

6) Line 495. Authors should check the meaning of “control patients”. Should it be “control children”?

7) Lines 499-501. It seems to me that a p value equal to .10 simply points to a non-significant difference.

8) The results section is quite long and difficult to follow. Maybe authors could insert a table reporting for each variable the group, time, and interaction effects, highlighting the significant effects in bold or through symbols.  

Discussion

1) Lines 598-601. Authors should put forward some explanations of these discrepancies.

2) Lines 608-615 and lines 625-627. Please refer to my previous comment.

3) Lines 631-633. From my point of view this limitation is a major flaw.

4) Lines 637-638. Authors should check “we children from…”.

Reviewer 2 Report

This paper presents an interesting study on the effects of moderate intensity physical activity 
in attention-deficit hyperactivity disorder 
(ADHD) children. The study consisted in applying the KiTAP computer-based game to 3 groups: medicated children with ADHD, non-medicated children with ADHD, and typically developing children. Several characteristics were measured such as alertness, distractibility, divided attention, flexibility, and go/no-go. Experimental results give interesting insights on these parameters and suggest that the moderate intensity physical activity has a positive impact in ADHD children.

Overall, the paper addresses an interesting topic on children's health and exercise and health and is suitable for IJERPH. The paper is well written; it is easy to read and to follow. Experimental results are formally analyzed. I recommend addressing the following remarks prior to definitive acceptance:

A) Content (Major)

a.1) Being the KiTAP test a central topic in the paper. I recommend including a picture of the experimental setup.

a.2) All 11 figures are extremely low quality. They have to be improved.

B) Typos (Minor)

b.1) 52: sympathomimetic drugs as methylphenidate 
…

--> sympathomimetic drugs such as methylphenidate 
…

b.2) 146: For 
the purpose to motivate …

--> For 
the purpose of motivating …

b.3) 231: The exist…

--> the existence…

Reviewer 3 Report

The topic of the manuscript “Acute physical activity, executive function and attention performance in children with attention-deficit hyperactivity disorder and typically developing children” is very interesting issue for the IJERPH readers.

However, the authors must improve the organization of the information shown in order to allow the reader to better understand the interest of the study and its results. Moreover, it would be advisable to review the statistical analyses.

a) According to STROBE guidelines, the study design should be included in the title.

b) Abstract: The authors should specify whether the unmedicated children in their study have any diagnosis or, on the contrary, they are children with neurotypical development.

c) Abstract: Are the children who watch the cartoon video children from the same subgroups (medicated, unmedicated and neurotypical development) than the control group? This point is unclear.

d) Abstract: The authors are recommended to review the summary section corresponding to the methodology. They should remove information about the context and include aspects such as study design, instruments for measuring the main variables, and total sample size.

e) Introduction: The authors should review a larger bibliography in order to widely inform the reader about alternative treatments to pharmacological treatments that are proposed for children with ADHD, such as, for example, intervention based on occupational therapy based on sensory integration.

f) Introduction: It would be advisable to better clarify the study objective or objectives in order to be able to respond to them later.

g) Methods: The “Participants” section must be renamed to “Design and participants” and, at the beginning of this section, it must include the key elements of the study design.

h) Methods: It is highly recommended to provide relevant contextual information in this section, such as temporary recruitment and intervention periods. Please delete this data from line 263 (“Results”).

i) Methods: The section in which the participants are described is not clearly structured. It would be positive to specify the total sample size and the subgroups to be recruited. In this regard information as where the participants of each subgroup were obtained, their corresponding inclusion criteria and, how they are distributed in the intervention group or the control group must be included.

j) Methods: Are the medicated children all comparable to each other? It must be recommendable to indicate the type of medication received by medicated children and, if this type of medication is the same or simulate among them, making the subgroup comparable. In case any matching techniques have been used in the study, they must be mentioned.

k) Methods: The decision of choosing the age range (from 6 to 12 years), based on what is derived?

l) Measures: The authors must provide more information about the “MINI Kid” assessment tool, since its purpose is not clear. In case it is a psychiatric diagnostic tool, who administers this instrument in the study?

m) Measures: Given that in later sections it is indicated that children's physiological measurements are collected (blood pressure and heart rate), it must also be specified in the corresponding section of “Measures”.

n) Experimental protocols and procedures: The information about results shown in lines 213-214 should be moved to the "Results" section.

o) Statistical analysis: The part of statistical analyses carried out is confusing. In order to clarify this section, the following information must be included: what type of analysis and tests have been used to try to answer the queries raised in the study.

p) Statistical analysis: It would be advisable for authors to include basic information such as the level of significance that is established and the type of statistical tests used.

q) Results: The information in Table 2 must be reorganized and rewritten in a different way. The frequency and percentage of the categorical variables: n (%) must be indicated. In addition, it is advisable to reformulate the table title and indicate how the sociodemographic variables shown in these results have been collected (the authors can consult the following reference as an example):

Gonzalez-Palacios, S.; Navarrete-Muñoz, E.-M.; García-de-la-Hera, M.; Torres-Collado, L.; Santa-Marina, L.; Amiano, P.; Lopez-Espinosa, M.-J.; Tardon, A.; Riano-Galan, I.; Vrijheid, M.; Sunyer, J.; Vioque, J. Sugar-Containing Beverages Consumption and Obesity in Children Aged 4–5 Years in Spain: the INMA Study. Nutrients 2019, 11, 1772.

r) Results: The information volume in Tables 3 and 4 is very confusing and is it not relevant to the results obtained. It is highly recommendable to filter this information.

s) Results: The results are not understandable as well as very confusing. They must be directly related to the study objectives and the statistical analyses must be consistent with the above. This epigraph must be completely reorganized (please review this information of interest).

Altman DG, Gore SM, Gadner MJ, Pocock SJ. Statistical guidelines for contributors to medical journals. Br Med J 1983;286: 1.489-1.493

Bailar JC, Mosteller F. Guidelines for statistical reporting in articles for medical journals: amplifications and explanations. Ann Intern Med 1988;108: 266-273

t) Discussion: Both, internal validity, taking into account the magnitude and direction of possible biases, as well as the external validity, must be discussed in the study.

u) Discussion: The authors should include a brief section summarizing the implications of work for practice and research.

v) References: The bibliographic references must be reviewed. The Vancouver style does not include the "doi" as an element in the general outline of scientific articles.

Round 2

Reviewer 1 Report

Manuscript ID: ijerph-806811

Title: Acute physical activity, executive function and attention performance in children with attention-deficit hyperactivity disorder and typically developing children: an experimental study

Journal: International Journal of Environmental Research and Public Health

Introduction

1) Lines 99-100. From my point of view, authors should be more specific about the expected differences between physical activity and control conditions among the three groups.

Discussion

1) In their cover letter (reply number 3 about the discussion), authors wrote “We were not able to examine control group children in the morning, while they had to attend school at this time. However, medicated children with ADHD had to be examined in the morning, because they had to take their methylphenidate medication in the morning and the examination had to be carry out while the children were under the effect of medication (i.e. methylphenidate).

Furthermore, regarding both treatment naŃ—ve and medicated children with ADHD, we could not interrupt their therapeutic program, why they were during that week in Vadaskert Child Psychiatric Hospital and Outpatient Clinic”.

I suggest that they add this explanation within the manuscript.
